# Molecular and Functional Relevance of Na_V_1.8-Induced Atrial Arrhythmogenic Triggers in a Human *SCN10A* Knock-Out Stem Cell Model

**DOI:** 10.3390/ijms241210189

**Published:** 2023-06-15

**Authors:** Nico Hartmann, Maria Knierim, Wiebke Maurer, Nataliya Dybkova, Gerd Hasenfuß, Samuel Sossalla, Katrin Streckfuss-Bömeke

**Affiliations:** 1Clinic for Cardiology and Pneumology, University Medical Center, 37075 Göttingen, Germany; nico.hartmann@med.uni-goettingen.de (N.H.); wiebke.maurer@med.uni-goettingen.de (W.M.); ndybkov@med.uni-goettingen.de (N.D.); hasenfus@med.uni-goettingen.de (G.H.); samuel.sossalla@ukr.de (S.S.); 2DZHK (German Center for Cardiovascular Research), Partner Site Göttingen and Rhein Main, 61231 Bad Nauheim, Germany; maria.knierim@med.uni-goettingen.de; 3Clinic for Cardio-Thoracic and Vascular Surgery, University Medical Center, 37075 Göttingen, Germany; 4Departments of Cardiology at Kerckhoff Heart and Lung Center, Bad Nauheim and University of Giessen, 61231 Bad Nauheim, Germany; 5Institute of Pharmacology and Toxicology, University of Würzburg, 97078 Würzburg, Germany

**Keywords:** Na_V_1.8, iPSC-cardiomyocytes, late Na^+^ current (I_NaL_), CRISPR Cas9

## Abstract

In heart failure and atrial fibrillation, a persistent Na^+^ current (I_NaL_) exerts detrimental effects on cellular electrophysiology and can induce arrhythmias. We have recently shown that Na_V_1.8 contributes to arrhythmogenesis by inducing a I_NaL_. Genome-wide association studies indicate that mutations in the *SCN10A* gene (Na_V_1.8) are associated with increased risk for arrhythmias, Brugada syndrome, and sudden cardiac death. However, the mediation of these Na_V_1.8-related effects, whether through cardiac ganglia or cardiomyocytes, is still a subject of controversial discussion. We used CRISPR/Cas9 technology to generate homozygous atrial *SCN10A*-KO-iPSC-CMs. Ruptured-patch whole-cell patch-clamp was used to measure the I_NaL_ and action potential duration. Ca^2+^ measurements (Fluo 4-AM) were performed to analyze proarrhythmogenic diastolic SR Ca^2+^ leak. The I_NaL_ was significantly reduced in atrial *SCN10A* KO CMs as well as after specific pharmacological inhibition of Na_V_1.8. No effects on atrial APD_90_ were detected in any groups. Both *SCN10A* KO and specific blockers of Na_V_1.8 led to decreased Ca^2+^ spark frequency and a significant reduction of arrhythmogenic Ca^2+^ waves. Our experiments demonstrate that Na_V_1.8 contributes to I_NaL_ formation in human atrial CMs and that Na_V_1.8 inhibition modulates proarrhythmogenic triggers in human atrial CMs and therefore Na_V_1.8 could be a new target for antiarrhythmic strategies.

## 1. Introduction

Voltage-gated sodium channels (Na_V_) trigger the fast upstroke of the action potential (AP), making them important for the physiological conduction of electrical impulses in the heart. Under physiological conditions, Na_V_ channels (predominantly Na_V_1.5) quickly become inactive after activation. However, in some cardiac pathologies such as ischemia and heart failure (HF), Na_V_ channels were described to remain persistently open or reopen, thus creating the late sodium current (I_NaL_) as a persistent inward current [1,2,3,4]. It has been demonstrated that this pathologically enhanced I_NaL_ has detrimental effects on cellular electrophysiology and can induce arrhythmias [3,5,6,7,8]. Previous reports have been published on the existence of non-cardiac Na_V_ isoforms in the heart including Na_V_1.8. Na_V_1.8 is encoded by the *SCN10A* gene and described as a voltage-gated sodium channel like the predominant cardiac isoform Na_V_1.5. Na_V_1.8 has been shown to be predominantly expressed in neuronal tissues with mainly nociception functions and in human and rat spinal cord ganglia and cranial sensory ganglia [9,10,11,12]. Recent work has demonstrated that Na_V_1.8 mRNA is expressed in murine and human myocardia [13,14]. In situ hybridization experiments displayed that Na_V_1.8 has comparable cellular localizations to Na_V_1.5 in murine cardiomyocytes [15]. Genome-wide association studies reported that variants in the *SCN10A* gene (coding for Na_V_1.8) are associated with cardiac arrhythmias such as atrial fibrillation, sudden cardiac death [16,17], impaired conduction in the form of alterations in the PQ and QRS intervals, heart rate and increased arrhythmogenic risk [16], and with J-wave syndromes, specifically Brugada syndrome (BrS) and early repolarization syndrome (ERS) [18]. However, it remains controversial whether these Na_V_1.8-associated effects are mechanistically mediated by Na_V_1.8 and, if so, if they occur in cardiac ganglia or cardiomyocytes (CMs). Na_V_1.8 mRNA and protein were found to be significantly more abundant in human atrial myocardium compared to the ventricular myocardium. The expression levels of Na_V_1.8 and Na_V_1.5 did not show any differences between myocardial samples obtained from patients with atrial fibrillation and those with sinus rhythm [19,20,21,22]. Functional single-cell experiments of atrial and ventricular human and murine CMs demonstrated direct effects of pharmacological Na_V_1.8 inhibition on the I_NaL_ and cellular arrhythmogenesis [19,22]. However, these studies were limited by utilizing respective ion channel blockers that could theoretically have unspecific effects. The currently available drugs, such as amiodarone, have limited efficacy, poor tolerability, and notable adverse side effects, including life-threatening ventricular arrhythmias. Clinical guidelines recommend amiodarone treatment for most patients with severe structural heart disease, especially heart failure (HF). However, chronic use of amiodarone can lead to severe extra-cardiac side effects and organ toxicity despite its relative effectiveness against arrhythmias. There is a demand for new and safer innovative compounds to address this issue. Therefore, we aimed to investigate the electrophysiological contribution of Na_V_1.8 using CRISPR/Cas9-generated homozygous atrial *SCN10A* knock out (KO) induced-pluripotent stem cell CMs (iPSC-CMs). We ultimately describe the influence of the Na_V_1.8 channel on the electrophysiological and molecular properties of human atrial CMs and further demonstrate that Na_V_1.8 is a potential new target for atrial antiarrhythmic strategies.

## 2. Results

### 2.1. CRISPR/Cas9 Based Homozygous Knock-Out of SCN10A in Human Atrial iPSC-Cardiomyocytes

Homozygous *SCN10A*/Na_V_1.8-deficient (*SCN10A* KO) human atrial iPSC-CMs were generated using CRISPR/Cas9 genome editing as previously described [22,23]. Full pluripotency, genome integrity and spontaneous differentiation capacity into all three germ layers were confirmed in *SCN10A* KO iPSCs [23]. Homozygous KO of the gene was confirmed by Sanger sequencing in two different *SCN10A*-KO iPSC cell lines [22,23] as well as in atrial differentiated *SCN10A* KO iPSC-CMs by showing premature stop codons on both alleles (A1: delC/insCAC → premature stop in Ex1 and A2: delCT → premature stop in Ex1) (Figure 1a). Successful differentiation of control and *SCN10A* KO iPSCs into atrial iPSC-CMs was demonstrated by immunostaining of atrial myosin light chain 2 isoform (MLC2a) (Figure 1b) and mRNA expression of the atrial marker PITX2 (Figure 1c).

### 2.2. Influence of Na_V_1.8 on I_NaL_ in Human Atrial iPSC-Cardiomyocytes

We hypothesized that the KO of *SCN10A* in atrial iPSC-CMs would reduce the proarrhythmogenic I_NaL_. Therefore, whole-cell voltage clamp experiments were performed to direct measure the I_NaL_ integral in human atrial *SCN10A* KO and control iPSC-CMs.

Since the amplitude of the I_NaL_ is relatively small in healthy hiPSC-CMs under physiological conditions [24], we used isoproterenol (Iso, 50 nmol/L) for slight beta-adrenergic stimulation in control and experimental groups during all functional experiments as described previously [20]. To further compare *SCN10A* KO with pharmacological inhibition of Na_V_1.8 and test for potential side effects of either KO or pharmacological intervention, we used the specific Na_V_1.8 blocker PF-01247324 (1 µmol/L) [19,22]. Voltage-clamp experiments demonstrated that the I_NaL_ was significantly reduced by genetical KO of Na_V_1.8 as well as by pharmacological inhibition. The Iso-induce increase in the I_NaL_ in control iPSC-CMs (−125.5 ± 8.4 A*ms*F^−1^) was significantly reduced in KO iPSC-CMs (−34.4 ± 4.8 A*ms*F^−1^, *p* < 0.0001, Figure 2). Moreover, the I_NaL_ was reduced to the level of KO iPSC-CMs by application of the specific Na_V_1.8 inhibitor [PF-01247324, 1 µmol/L, atrial control iPSC-CMs vs. PF-01247324 (−44.9 ± 3.8 A*ms*F^−1^, *p* < 0.001, Figure 2)]. Notably, we observed no additional effects on the I_NaL_ in KO iPSC-CMs after application of PF-01247324.

### 2.3. Effects of Na_V_1.8 on the Atrial Action Potential

To assess the potential influence of KO and pharmacological Na_V_1.8 inhibition on the action potential characteristics in human atrial iPSC-CMs, we performed whole-cell current-clamp experiments. The data presented herein are representative of measurements conducted at a frequency of 1 Hz. No effects on atrial action potential duration at 90% repolarization (APD_90_) were observed in KO iPSC-CMs as well as after the additional application of the specific Na_V_1.8 blocker PF-01247324 (Figure 3a,b; control at 1.0 Hz, 243.0 ± 30.5 ms vs. control + PF 229.3 ± 19.0 ms,−5.8%; KO control 204.4 ± 24.2 ms, −16%, vs. KO + PF 198.9 ± 30.1 ms, −3%). Furthermore, no discernible impacts were observed on the duration of atrial action potential at 20% repolarization (APD_20_), action potential duration at 50% repolarization (APD_50_), and action potential duration at 70% repolarization (APD_70_). The available data, including Appendix A, were included in the Appendix A. To rule out potential side effects of KO or pharmacological inhibition of Na_V_1.8, we compared the resting membrane potential and action potential amplitude in all groups. No significant effects of KO or pharmacological inhibition of Na_V_1.8 on either AP amplitude (APA, Figure 3c, 113.7 ± 4.4 ms vs. control + PF 118.7 ± 3.4 ms, KO control 105.7 ± 5.1 ms, KO + PF 102.1 ± 4.2 ms ), resting membrane potential (RMP, Figure 3d; −76.0 ± 6.2 ms vs. control + PF −72.2 ± 5.7 ms; KO control −66.7 ± 2.6 ms vs. KO + PF −67.3 ± 3.9 ms), or upstroke velocity (Vmax, Figure 3e; 106.2 ± 11.2 vs. control + PF 127.5 ± 11.9 mV/ms; KO control 92.7 ± 13.8 vs. KO + PF 82.8 ± 12.4 mV/ms) could be observed.

### 2.4. Effects of Na_V_1.8 on Atrial Sarcoplasmic Reticulum Ca^2+^ Leak and Arrhythmogenesis

As we previously demonstrated, Na_V_1.8 exerts its arrhythmogenic potential in the atria via enhancement of the I_NaL_ [19]. To investigate the functional cellular effects of the Na_V_1.8-dependent I_NaL_ on Ca^2+^ homeostasis and cellular arrhythmogenesis in atrial iPSC-CMs, we recorded line scans in confocal microscopy experiments using Fluo 4-AM in human atrial *SCN10A* KO and control iPSC-CMs. Diastolic confocal line scans (Fluo 4-AM) showed that KO of *SCN10A* in atrial iPSC-CMs massively decreased the frequency of spontaneous arrhythmogenic Ca^2+^ sparks compared to the respective control cells (KO: 3.25 ± 0.23 sparks/100 µm/s vs. control: 6.34 ± 0.43, *p* = 0.0142). Similarly, pharmacological inhibition of Na_V_1.8 by PF-01247324 led to a significant reduction of diastolic Ca^2+^ sparks in atrial control iPSC CMs (control + PF-01247324: 3.96 ± 0.21, *p* = 0.0469), while having no further effect on *SCN10A* KO cells (KO + PF-01247324: 3.47 ± 0.34, *p* = 0.9998) (Figure 4a,b). Furthermore, we investigated the incidence of spontaneous diastolic Ca^2+^ waves as major arrhythmogenic events. The proportion of cells exhibiting diastolic Ca^2+^ waves was significantly reduced from 24.7% in atrial control -iPSC-CMs to 5.5% in the *SCN10A* KO group. After pharmacological inhibition of Na_V_1.8, we observed a comparable reduction of cells displaying Ca^2+^ waves compared to control (9.0%). There was no significant additional effect of Na_V_1.8 inhibition in *SCN10A* KO cells (8.7%) (Figure 4c,d).

### 2.5. Influence of SCN10A KO on Intracellular Ca^2+^ Transients

Since KO of *SCN10A* was shown to reduce the arrhythmogenic potential of the increased I_NaL_ in atrial human iPSC-CMs by reduction of spontaneous SR Ca^2+^ release events, we further sought to rule out any potential adverse effects on cellular Ca^2+^ handling.

We therefore performed epifluorescence microscopy (Fura 2-AM) in atrial iPSC-CMs with and without KO of *SCN10A* and/or pharmacological inhibition of Na_V_1.8. *SCN10A* KO did not show any significant effects on Ca^2+^ transient amplitude, diastolic Ca^2+^ levels, time to peak, or relaxation time (RT 80%), demonstrating intact Ca^2+^ handling in both KO and WT cells. Of note, specific inhibition of Na_V_1.8 by PF-01247324 also did not exert any additional effects on Ca^2+^ transient parameters in either control or KO atrial iPSC-CMs (Figure 5).

### 2.6. The Expression of Key Proteins of Excitation–Contraction Coupling Is Not Altered by a SCN10A KO

Since we demonstrated a reduction in the arrhythmogenic potential in atrial human *SCN10A* KO iPSC-CMs, we wanted to analyze the potential underlying effects on a molecular level. Therefore, we investigated the expression of key proteins of excitation–contraction coupling (voltage-gated sodium channel isoform Na_V_1.5; L-type Ca^2+^ channel Ca_V_1.2; cardiac ryanodine receptor 2, RyR_2_) using Western blot experiments.

In atrial control iPSC-CMs, we found a lower expression of Na_V_1.5 compared to SCN10A KO iPSC-CMs, but it did not reach statistical significance (Figure 6a,d). Furthermore, RyR2 and Ca_V_1.2 were not regulated in atrial *SCN10A* KO iPSC-CMs compared to control atrial iPSC-CMs according to the Western blot results (Figure 6b,c,e,f). Thus, *SCN10A* KO seems to exert no significant side effects on the expression of the other main proteins relevant to excitation–contraction coupling in atrial iPSC-CMs compared with their respective control cells.

## 3. Discussion

Atrial fibrillation (AF) is the most prevalent clinically significant arrhythmia. It represents a major risk factor for embolic stroke and exacerbation of heart failure (HF), consequently contributing to heightened morbidity and mortality rates [25]. The current prevalence of atrial fibrillation (AF) in adults ranges between 2% and 4%, with an anticipated 2.3-fold increase due to the extended longevity of the general population. For patients with atrial fibrillation (AF), first-line therapies for rhythm control include anti-arrhythmic drugs and/or left atrial pulmonary vein ablation [25]. However, pharmacological rhythm control is notably restricted in patients with underlying structural heart disease. The currently available drugs for these patients have limitations, poor tolerability, and adverse side effects [26]. Therefore, there is a demand for new and safer innovative compounds to address the treatment of AF in patients with structural heart disease. Sodium currents are effective therapeutic targets for the treatment of AF. In this context, the I_NaL_ has been increasingly identified as a potential target to inhibit cellular arrhythmogenic triggers in AF and the first hopeful results have been shown in clinical trials [26,27,28,29,30].

However, the mechanism of I_NaL_ regulation with respect to cellular arrhythmogenic triggers is not yet well understood. Besides Na_V_1.5, other Na_V_ isoforms have been reported to be present in the heart. We have shown that the expression of the Na^+^ channel Na_V_1.8 in left ventricular CMs is upregulated in human HF myocardium [20], and that Na_V_1.8 contributes to arrhythmogenesis by inducing the I_NaL_ [19,20,22,31]. Variants in the *SCN10A* gene (Na_V_1.8) were shown to be associated with cardiac arrhythmias such as atrial fibrillation and sudden cardiac death [32]. Whether these Na_V_1.8-related effects are mediated by cardiac ganglia or cardiomyocytes is still under debate. In the present study, we used human atrial *SCN10A* KO iPSC-CMs and demonstrated that Na_V_1.8 is responsible for the generation of the I_NaL_. Both inhibition and KO of Na_V_1.8 potently suppressed the I_NaL_ and diastolic SR-Ca^2+^ leak as proarrhythmogenic triggers in atrial CMs. These findings suggest that targeting Na_V_1.8 constitutes a novel therapeutic antiarrhythmic strategy for the treatment of atrial rhythm disorders.

### 3.1. Na_V_1.8 and Atrial I_NaL_

Under pathological conditions, the enhanced persistent Na^+^ influx, known as enhanced I_NaL_, has been demonstrated to play an important role throughout the action potential [33]. The prolongation of the action potential duration caused by an I_NaL_ increases the likelihood of early afterdepolarizations (EADs), which serve as triggers for the occurrence of arrhythmias. The specific Na_V_ isoforms involved in the generation of an I_NaL_, particularly in clinically relevant conditions like atrial fibrillation (AF) and heart failure (HF), remain unclear. This information is of translational relevance because selectively targeting the inhibition of the I_NaL_ would be a desirable antiarrhythmic approach.

Genome-wide association studies have identified *SCN10A* as a regulator of cardiac conduction. By employing various methodologies in both human and mouse cardiomyocytes, we have demonstrated the significance of Na_V_1.8 in the generation of the late sodium current (I_NaL_). We found that Na_V_1.8 is upregulated under conditions of HF and cardiac hypertrophy [20,22,31]. Recent studies have provided evidence for the involvement of Na_V_1.8 in atrial cellular electrophysiology and have successfully linked *SCN10A* variants to AF [32,34]. However, some of the preliminary studies are limited by the use of appropriate ion channel blockers, which theoretically could have nonspecific effects.

Therefore, in the present study we used homozygous atrial *SCN10A*-KO iPSC-CMs to show that the Na_V_1.8-associated effects are mechanistically mediated by Na_V_1.8. Since under healthy conditions the I_NaL_ is very low, we applied isoproterenol in order to enhance the I_NaL_ for a better comparison between the control and KO iPSC CMs. Casini et al. did not detect any Na_V_1.8-based I_NaL_ in non-diseased human atrial and rabbit ventricular CMs without beta-adrenergic stimulation [24]. Most importantly, the incidence of an enhanced I_NaL_ depends on pharmacological (beta-adrenergic activation) or pathological stimulation and explain the absence of Na_V_1.8 effects in this study. Here, we show that Na_V_1.8 contributes to an enhanced I_NaL_ in atrial control iPSC-CMs by reducing the I_NaL_ by simultaneous treatment with isoproterenol and PF-01247324. Moreover, the specific blocker PF-01247324, when used to inhibit Na_V_1.8, did not induce any additional effects on the I_NaL_ in Na_V_1.8 KO atrial cells compared to untreated Na_V_1.8 KO atrial cells. This finding highlights the specificity of the drug in targeting Na_V_1.8 [35]. Pabel et al. demonstrated that both pharmacological inhibition and genetic ablation of Na_V_1.8 resulted in a reduction of the late sodium current (I_NaL_) in human and murine atrial CMs [19]. In line with this, patch-clamp recordings of isolated human atrial CMs obtained from patients in sinus rhythm revealed that following mild beta-adrenergic stimulation with isoproterenol, the inhibition of Na_V_1.8 using PF-01247324 and A-803467 led to a significant reduction in the late sodium current (I_NaL_) [19]. Isolated atrial CMs from *SCN10A*-/- mice revealed a significantly lower I_NaL_ compared to WT while pharmacological inhibition by PF-01247324 exerted no additional effect on the I_NaL_ in *SCN10A*-/- mice [19]. Therefore, the results of the present study are in line with previous findings in atrial human and mice atrial KO [19] and ventricular KO mice and human iPSC-CMs and isolated CMs [22,36]. Moreover, the impact of *SCN10A* variants associated with AF on the modulation of the I_NaL_ was demonstrated through transfection experiments in ND7/23 cells. This additional evidence further strengthens the notion that Na_V_1.8 plays a significant role in the development of I_NaL_-related arrhythmias [37].

### 3.2. Na_V_1.8 and Atrial Action Potential Duration

Previous studies have provided evidence that the I_NaL_ plays a significant role in determining the APD in both atrial and ventricular CMs [2,3,8,27,29]. Having demonstrated the upregulation of Na_V_1.8 expression in human AF and HF, we proceeded to investigate the impact of Na_V_1.8-induced I_NaL_ on various action potential parameters. We also used isoproterenol to enhance the I_NaL_. In line with previous data from our group in atrial human and mice CMs [19], the present study showed that in atrial control or *SCN10A* KO iPSC CMs, Na_V_1.8 has negligible effects on the atrial action potential parameters. In AF, the APD becomes shorter and a further shortening of APD may lead to shorter refractory periods, thereby further facilitating reentry. Therefore, negligible effects on APD point towards a positive therapeutic profile of targeting Na_V_1.8 in AF. Since dv/dt is a surrogate for the fast Na^+^ influx and peak Na^+^ current, these data show that there is no involvement of Na_V_1.8 in the peak Na^+^ current in atrial iPSC CMs. A negligible effect of Nav1.8 inhibition on cardiac conduction peak Na^+^ current blockade would be desirable in order to treat patients with structural heart disease and AF.

### 3.3. Na_V_1.8 and Atrial Ca^2+^ Handling

In our previous studies, we demonstrated that the I_NaL_-mediated Na^+^ influx has the ability to induce Ca^2+^ influx through reverse-mode NCX, resulting in elevated cytosolic [Ca^2+^] levels and an increased occurrence of Ca^2+^ sparks in the human atrium [21,28]. Furthermore, the inhibition of the I_NaL_ through specific targeting of Na_V_1.8 has the capability to reduce the reverse mode NCX, thereby also mitigating diastolic proarrhythmogenic SR-Ca^2+^ leak [20,21,31]. The relationship between enhanced I_NaL_ and an increased risk of arrhythmias is indeed complex. This complexity arises from the fact that the increased leak of Ca^2+^ from the SR can induce a transient inward current, which, in turn, leads to arrhythmogenic delayed afterdepolarizations. Additionally, it can also result in significant spontaneous proarrhythmic Ca^2+^ release from the SR [8,28].

In the present study, we show a reduction in spontaneous SR Ca^2+^ spark frequency as well as a decreased frequency of spontaneous Ca^2+^ waves in human atrial *SCN10A* KO CMs and in control CMs after pharmacological inhibition. As Ca^2+^ waves represent a major proarrhythmic trigger, we hereby establish the principle of Na_V_1.8-induced I_NaL_ and its triggering role in cellular arrhythmogenesis that is independent of neuronal influence in isolated human atrial CMs. Interestingly, we found no effects on intracellular Ca^2+^ transients in either *SCN10A* KO or following Na_V_1.8 inhibition in control CMs. Thus, we propose that the intracellular Ca^2+^ handling and likely contractile function of CMs remain mostly unaffected by Na_V_1.8. In summary, our results and current evidence indicate that the discussed Na_V_1.8-induced I_NaL_ mainly influences arrhythmogenesis on a subcellular level while leaving cellular Ca^2+^ release and contractile function unaffected.

### 3.4. Clinical Relevance

The currently available anti-arrhythmic drugs, particularly for patients with structural heart disease, are limited in their effectiveness. Drugs like flecainide or amiodarone, which are commonly used, demonstrate suboptimal efficacy and are associated with significant adverse side effects, including life-threatening ventricular arrhythmias and organ toxicity. Therefore, new, safer, and more precise compounds for the treatment of atrial arrhythmias are highly desirable. Na_V_1.8 was detected in atria, and human hypertrophied and failing ventricles [19,22,31]. The results of the present study demonstrate that either genetic ablation of Na_V_1.8 using *SCN10A* KO iPSC-CMs or pharmacological inhibition can reverse cellular proarrhythmic effects in the atria. Both inhibition and KO of Na_V_1.8 potently suppressed proarrhythmogenic triggers (e.g., I_NaL_ and diastolic SR-Ca^2+^ leak) while leaving the peak Na^+^ current unaffected. These findings suggest targeting Na_V_1.8-dependent I_NaL_ constitutes a novel therapeutic anti-arhythmic strategy for the treatment of atrial rhythm disorders.

## 4. Materials and Methods

### 4.1. Generation of Homozygous Knockout iPSCs Using CRISPR/Cas9 and Directed Differentiation into Atrial iPSC-Cardiomyocytes

All procedures conducted in this study adhered to the principles outlined in the Declaration of Helsinki and received approval from the local ethics committee of the University Medicine of Göttingen (Az-10/9/15). Informed consent was signed by all tissue donors. A homozygous *SCN10A* KO iPSC line was generated from a control iPSC line by CRISPR/Cas9 genome editing as described in detail in previous studies [22,23]. The generated *SCN10A* KO iPSCs were differentiated into functionally beating, atrial iPSC-derived cardiomyocytes as described in [38].

In order to achieve directed atrial cardiac differentiation of the induced pluripotent stem cells (iPSCs), manipulation of the Wnt signaling pathway was employed, as previously described [38]. The cells were cultured for 60 days and then passaged onto glass-bottom Fluoro Dishes (WPI, 30 K/dish) by subjecting them to trypsinization at 37  °C for 3 min. The cells were allowed to settle for 7 days prior to further measurements, with medium changes performed every 2 days. iPSC-derived cardiac myocytes (iPSC-CMs) were analyzed 8–10 weeks after the initiation of differentiation, unless otherwise specified. The purity of the iPSC-CMs was determined by flow analysis, with a focus on cardiac troponin T positivity (>90% cardiac TNT+), as well as through qPCR and immunofluorescence analysis of atrial-specific markers (PITX2, MLC2a). Four to five differentiation experiments were performed to generate atrial iPSC-CMs from two Na_V_1.8 knockout lines and their corresponding healthy isogenic control line.

### 4.2. Pharmacological Intervention

For selective inhibition of Na_V_1.8-induced sodium currents, a specific Na_V_1.8 blocker PF-01247324 (1 µmol/L, Sigma-Aldrich, Taufkirchen, Germany)) was used. Cellular electrophysiological measurements were performed under slight beta-adrenergic stimulation (isoproterenol (Iso), 50 nmol/L, Sigma-Aldrich, Taufkirchen, Germany)) [20]. Prior to the start of experiments, the CMs were incubated for 15 min with both substances or isoproterenol alone as a control.

### 4.3. Patch-Clamp Experiments

The patch-clamp experiments was performed as previously described [19,22]. Briefly, 35,000 atrial iPSC-CMs were plated on glass-bottom Fluoro Dishes and incubated with either isoproterenol (50 nmol/L, Sigma-Aldrich, Taufkirchen, Germany)) or isoproterenol + PF01247324 (1 µmol/L, Sigma-Aldrich, Taufkirchen, Germany)) for 15 min before starting the measurements. The experiments were conducted at room temperature.

Action potential recordings were performed using the whole-cell patch-clamp technique. To elicit action potentials, square current pulses with amplitudes of 0.5–1 nA and durations of 1–5 ms were applied. The stimulation frequency was increased gradually from 0.5 to 2 Hz.

The late sodium current (I_NaL_) was measured using the ruptured-patch whole-cell patch-clamp technique. The pipette used had a resistance ranging from 2 to 3 mega-ohms (MΩ). I_NaL_ recordings were performed exclusively in CMs where a seal with a resistance of over 1 giga-ohm (GΩ) was achieved, and the access resistance remained below 7 MΩ. After a stabilization period of 3 min, the iPSC-derived CMs were held at a holding potential of −120 mV and then depolarized to −35 mV for 1000 ms with 10 pulses and a basic cycle length of 2 s. The I_NaL_ was quantified as the integral current amplitude between 100 and 500 ms and was normalized to the membrane capacitance.

### 4.4. Confocal Ca^2+^ Imaging

A total of 35.000 atrial iPSC-CMs plated on glass-bottom FluoroDishes were incubated with the Ca^2+^ indicator Fluo 4-AM (10 µmol/L, Invitrogen, Darmstadt, Germany) for 15 min at RT for de-esterification of the dye. The solution was substituted with Tyrode’s solution (as described in [19]) and the respective pharmacological agents and left to incubate for 15 min. Confocal line scans were obtained with a laser scanning confocal microscope (LSM 5 Pascal, Zeiss, Jena, Germany). Scans were conducted after continuous electrical field stimulation at 1 Hz during pausing of stimulation. Ca^2+^ release events were analyzed using the SparkMaster plugin for ImageJ. The mean Ca^2+^ spark frequency was calculated from the number of sparks normalized to scan width, duration, and scan rate (100 µm/s). Cells exhibiting major Ca^2+^ release events (Ca^2+^ wavelets or waves) were excluded from the calculation of Ca^2+^ spark frequency and separately classified as proarrhythmic cells as a proportion of all cells.

### 4.5. Epifluorescence Microscopy for Ca^2+^ Transient Measurements

A total of 35.000 atrial CMs were dissociated and plated as described above and loaded with the radiometric Ca^2+^ indicator Fura 2-AM (5 µmol/L, Invitrogen) for 15 min at RT. Subsequently, the cells were washed with Tyrode’s solution for de-esterification and incubated with pharmacological agents as described above. The measurements were performed using a fluorescence detection system (IonOptix, Amsterdam, Netherlands) connected to an inverted microscope with oil immersion lens (40×). Cardiomyocytes were subjected to electrical field stimulation at 1 Hz for the duration of the experiment to ensure steady intracellular Ca^2+^ concentrations. Recording of Ca^2+^ transients for analysis was performed at 1 Hz at steady state. For each cell, the stimulation was paused for 30 s to detect spontaneous Ca^2+^ release events and evaluate the spontaneous beating frequency of the iPSC-CMs. Ca^2+^ transients were analyzed using the software IonWizard (IonOptix).

### 4.6. Statistical Analysis

The data are reported as mean ± SEM, unless otherwise stated. Analysis was carried out with Prism 9 software (Graphpad, San Diego, CA, USA). For comparisons of two groups, unpaired Student’s t test was used in the case of parametric distribution of the data. Three or more groups including more than one differentiation experiment were compared using nested one-way ANOVA. The results were corrected for multiple comparisons by Sidak’s correction. Fisher’s exact test was used to statistically compare proportions. *p* values are two-sided and considered statistically significant if *p* < 0.05.

## 5. Conclusions

In conclusion, we showed that the neuronal sodium channel Na_V_1.8, which contributes to the I_NaL_ in the heart, is down-regulated in atrial *SCN10A*-KO iPSC-CMs and, importantly, contributes to I_NaL_ formation in human atrial CMs. Na_V_1.8 KO or the inhibition of Na_V_1.8 modulates proarrhythmogenic triggers such as I_NaL_ and diastolic SR-Ca^2+^ leak in human atrial CMs. Therefore, Na_V_1.8 might represent a novel treatment target for antiarrhythmic strategies.

## Figures and Tables

**Figure 1 ijms-24-10189-f001:**
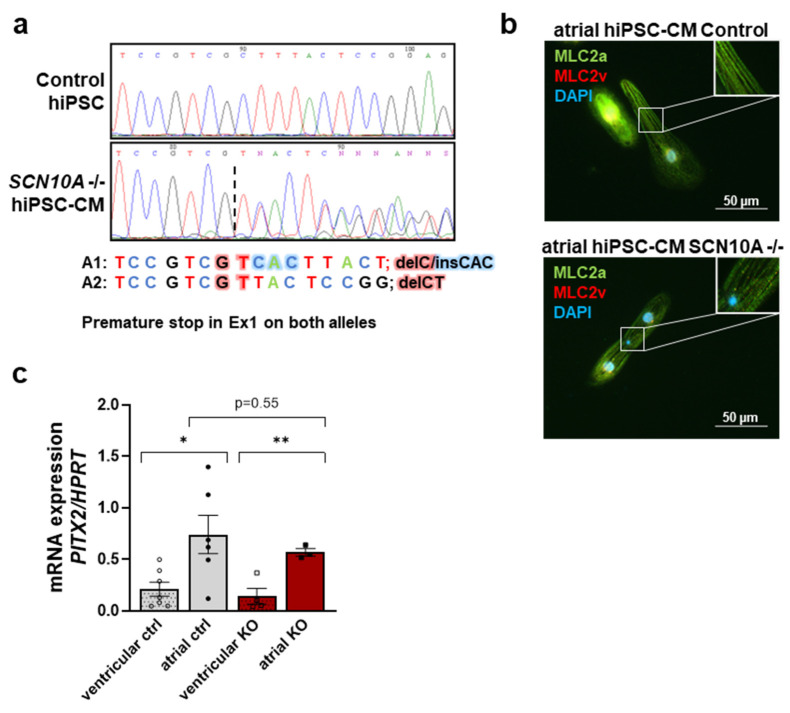
CRISPR/Cas9-based knock-out of *SCN10A*/Na_V_1.8 in atrial iPSC-CMs. (**a**) Sanger sequencing of control iPSCs and *SCN10A* iPSC-KO CMs demonstrating frameshifts in both alleles leading to a premature stop in exon 1 (A1: delC/insCAC and A2: delCT). (**b**) Atrial control and *SCN10A* KO iPSC-CMs were stained for MLC2a (green) and MLC2v (red) demonstrating atrial differentiation. Nuclei were stained with DAPI. (**c**) mRNA expression level of atrial marker PITX2 normalized to house-keeping gene HPRT in atrial control and *SCN10A* KO iPSC-CMs (*n* = 6/3 differentiations) compared to ventricular control and *SCN10A* KO iPSC-CMs (*n* = 7/4 differentiations). Student’s t-test was applied for normally distributed data. *: *p* < 0.05; **: *p* < 0.01.

**Figure 2 ijms-24-10189-f002:**
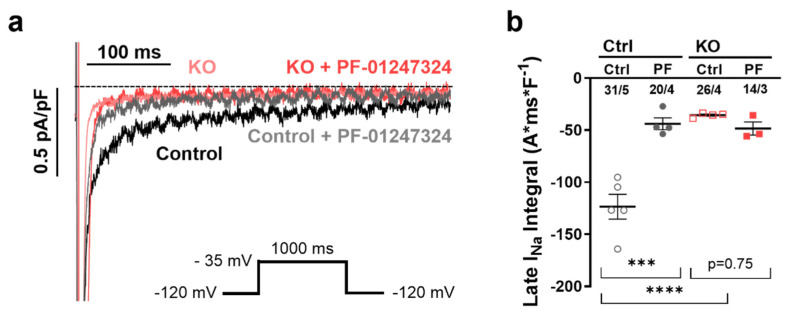
(**a**) Original traces of I_NaL_ in atrial control iPSC vs. *SCN10A* KO-iPSC cells according to the inserted protocol. (**b**) Mean values per differentiation ± SEM of I_NaL_ (atrial control *n* = 31 cells/5 differentiations; atrial control + PF *n* = 20 cells/4 differentiations; *SCN10A* KO iPSC−CM control *n* = 26 cells/4 differentiations, *SCN10A* KO-iPSC-CMs+ PF *n* = 14 cells/3 differentiations). Mean values per differentiation were compared using one-way ANOVA with Sidak’s test for multiple comparisons to calculate *p* values (*** = *p* < 0.001; **** = *p* < 0.0001).

**Figure 3 ijms-24-10189-f003:**
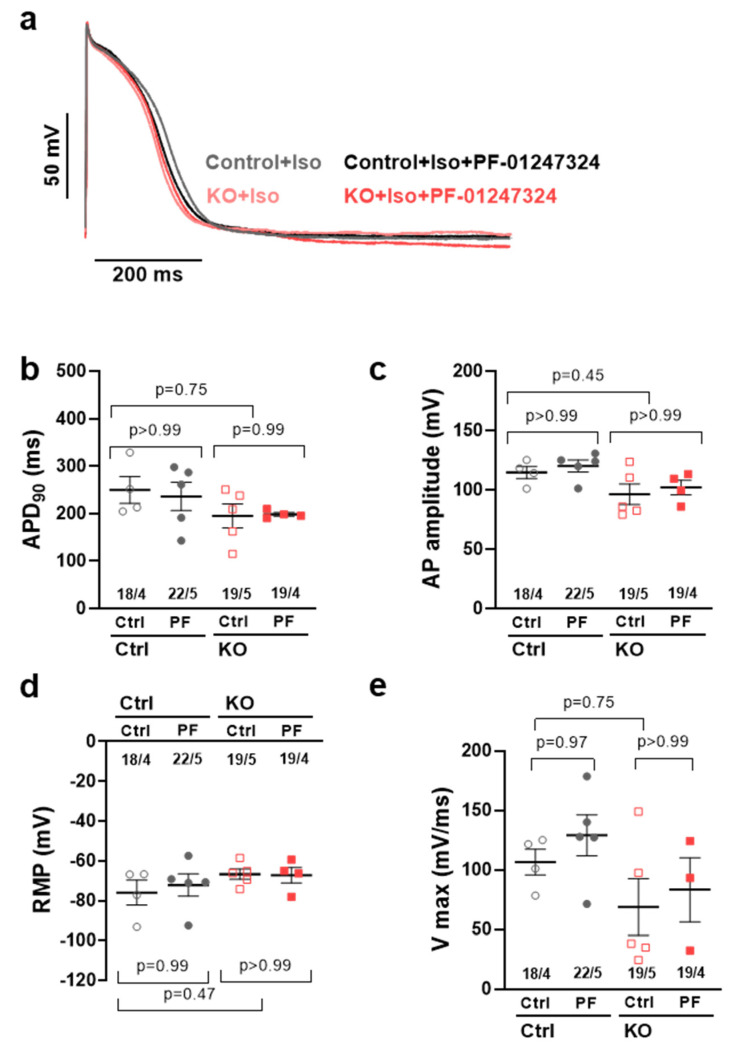
(**a**) Original traces of APD_90_ in atrial control iPSC−CMs vs *SCN10A* KO-iPSC-CMs at 1 Hz. (**b**) Mean (nested) ± SEM of APD_90_ (atrial control *n* = 18 cells/4 differentiations; atrial control + PF *n* = 22 cells/5 differentiations; *SCN10A* KO control *n* = 19 cells/5 differentiations; *SCN10A* KO + PF *n* = 19 cells/4 differentiations); statistics calculated using nested oneway ANOVA. (**c**) Mean ± SEM of amplitude (atrial control *n* = 18 cells/4 differentiations; atrial control + PF *n* = 22 cells/5 differentiations; *SCN10A* KO control *n* = 19 cells/5 differentiations; *SCN10A* KO + PF *n* = 19 cells/4 differentiations). (**d**) Mean ± SEM of RMP (atrial control *n* = 18 cells/4 differentiations; atrial control + PF *n* = 22 cells/5 differentiations; *SCN10A* KO control *n* = 19 cells/5 differentiations; *SCN10A* KO + PF *n* = 19 cells/4 differentiations); statistics calculated using nested one−way ANOVA. (**e**) Mean ± SEM of Vmax (atrial control *n* = 18 cells/4 differentiations; atrial control + PF *n* = 22 cells/5 differentiations; *SCN10A* KO control *n* = 19 cells/5 differentiations; *SCN10A* KO + PF *n* = 19 cells/4 differentiations).

**Figure 4 ijms-24-10189-f004:**
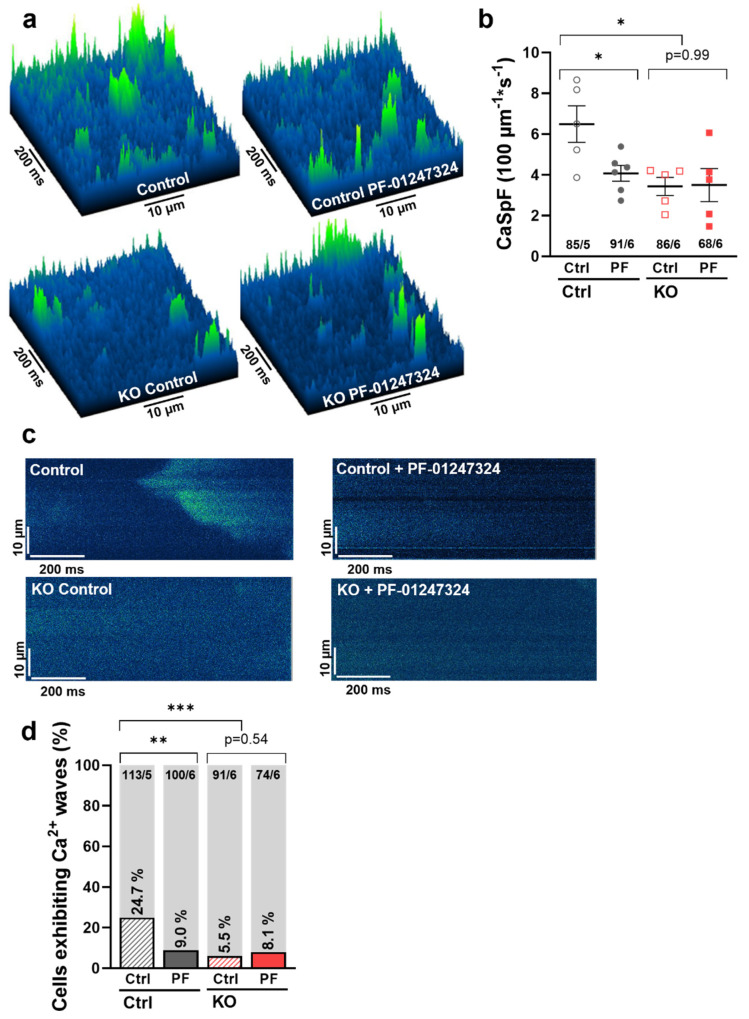
Contribution of Na_V_1.8 to spontaneous diastolic sarcoplasmic reticulum Ca^2+^ release in atrial iPSC-CMs. (**a**) Representative surface plots showing spontaneous diastolic Ca^2+^ sparks (green) in atrial iPSC-CMs. (**b**) Mean values of Ca^2+^ spark frequency (CaSpF) normalized to scan width and duration. Numbers indicate total cell count of control CMs (atrial control, *n* = 85 cells/5 differentiations), control cells treated with Na_V_1.8 inhibitor PF01247324 (atrial control + PF-01247324, *n* = 91/6), and atrial CMs with KO of Na_V_1.8 after control treatment and treatment with PF-01247324 (*n* = 86/6 vs. 68/6). Symbols indicate the mean values of different differentiation experiments. (**c**) Original representative line scans of atrial iPSC-CMs illustrating a spontaneous proarrhythmogenic diastolic Ca^2+^ wave (green). (**d**) Percentage of cells exhibiting diastolic Ca^2+^ waves in relation to cells without Ca^2+^ waves (grey bars) in atrial control CMs (24.7%, *n* = 28 of 113 cells from 5 differentiations) compared to atrial control CMs + PF-01247324 (9%, *n*= 9/100 cells/6 diff.) and to atrial SCN10A KO CMs with inhibition of Na_V_1.8 by PF-01247324 (8.1%, *n* = 6/74cells/6 diff.) or without (5.5%, *n* = 5/91 cells/6 diff.). Values are presented as mean ± SEM or absolute numbers. Mean values per differentiation were compared using one-way ANOVA with Sidak’s test for multiple comparisons to calculate *p* values. Proportions were compared using Fisher’s exact test (* = *p* < 0.05, ** = *p* < 0.01; *** = *p* < 0.001).

**Figure 5 ijms-24-10189-f005:**
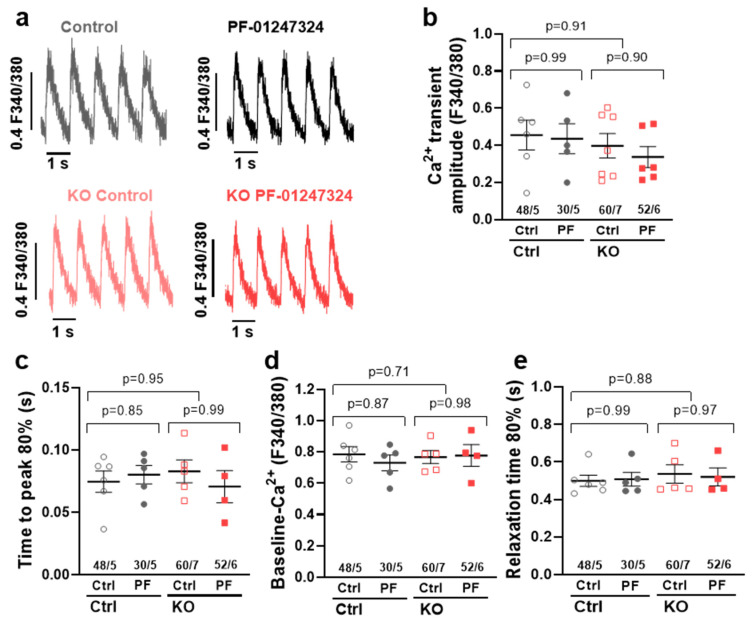
(**a**) Representative original recordings of stimulated systolic Ca^2+^ transients (epifluorescence microscopy, Fura 2-AM, 1 Hz) of human atrial *SCN10A* or control iPSC-CMs and after additional Na_V_1.8 inhibition by PF-01247324. Mean values ± SEM of (**b**) systolic Ca^2+^ transient amplitude, (**c**) time to peak 80%, (**d**) diastolic Ca^2+^ level, and (**e**) relaxation time 80% in control CMs (*n* = 48 cells/5 differentiations), *SCN10A* KO CMs (*n* = 60/7), and each after treatment with PF-01247324 (control + PF-01247324 *n* = 30/5, KO + PF-01247324 *n* = 52/6). Values are presented as mean ± SEM. Mean values per differentiation were compared using one-way ANOVA with Sidak’s test for multiple comparisons to calculate *p* values.

**Figure 6 ijms-24-10189-f006:**
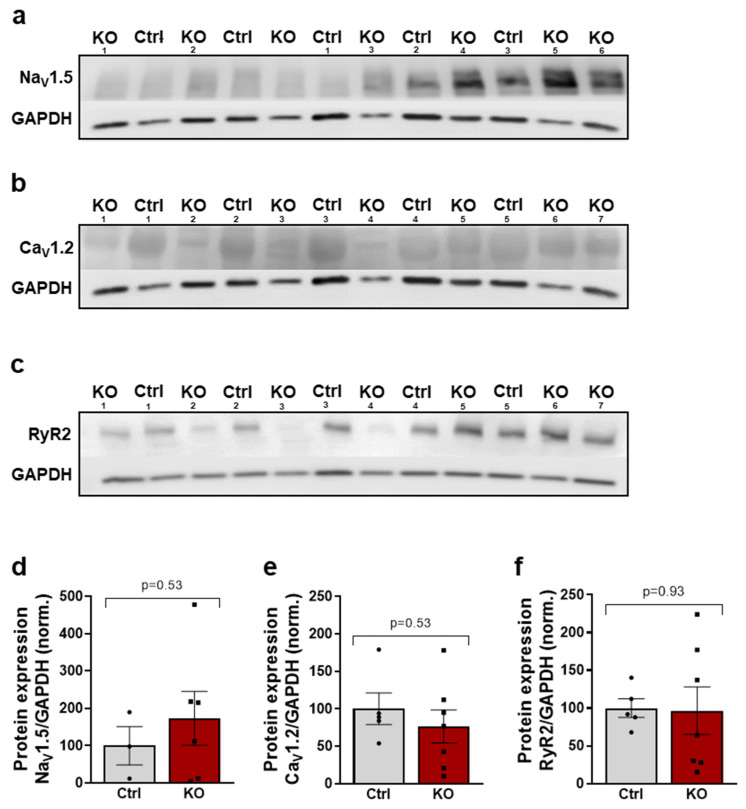
Original Western blots of Na_V_1.5 (**a**), Ca_V_1.2 (**b**), and RyR2 (**c**) in atrial control and *SCN10A* KO iPSC-CMs. Normalized values of Na_V_1.5 (**d**) (*n* = 3 control/6 KO differentiations), Ca_V_1.2 (**e**) (*n* = 5/7 differentiations), and RyR2 (**f**) (*n* = 5/7 differentiations) in atrial control and *SCN10A* KO iPSC-CMs normalized to GAPDH (*n* = 5/7 differentiations). Student’s t-test was used for statistical analysis.

## Data Availability

Not applicable.

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
