# Peer review of "Molecular and Functional Relevance of NaV1.8-Induced Atrial Arrhythmogenic Triggers in a Human SCN10A Knock-Out Stem Cell Model"

_ijms, 2023, doi:10.3390/ijms241210189_

Round 1

Reviewer 1 Report

The current study utilized SCN10A knockout atrial iPSC-CMs to investigate the role of the Nav1.8 isoform (encoded by SCN10A gene) in the electrophysiological properties of CMs and assess the potential of it inhibition as a clinical target for the treatment of atrial arrhythmias.

Authors found that genetic knockout and pharmacological inhibition of NaV1.8 significantly reduced INaL in atrial iPSC-CM, and the specific NaV1.8 inhibitor reduced INaL in control iPSC-CM to the level of KO iPSC-CM. Whole-cell current-clamp experiments were performed to evaluate the effects of genetic KO and pharmacological inhibition of NaV1.8 on action potential parameters in human atrial iPSC-CM, and no significant effects were observed on action potential duration, resting membrane potential, or action potential amplitude. The study found that both knockout of SCN10A and pharmacological inhibition of NaV1.8 led to a decrease in spontaneous arrhythmogenic Ca2+ sparks and waves in human atrial iPSC-CM, indicating a potential role of NaV1.8-dependent INaL in cellular arrhythmogenesis. Fura 2-AM calcium measurements demonstrated intact Ca2+ handling in both atrial iPSC-CM with and without KO of SCN10A and/or pharmacological inhibition of NaV1.8, with no significant effects on Ca2+ transient parameters.

Despite the clear relevance of the topic, the current study lacks originality and novelty and therefore cannot be recommended for publication. The authors used to investigate extensively the role of Nav1.8 in various models, including human heart failure atrial CMs (Dybkova, et al., 2018 Cardiovascular Research https://doi.org/10.1093/cvr/cvy152), wild type and SCN10A knockout murine CMs, and wild type and SCN10A knockout human ventricular iPSC-CMs (Bengel et al., 2021 Nat Commun https://doi.org/10.1038/s41467-021-26690-1). The present study uses previously generated SCN10A knockout iPSC lines (Bengel et al., 2021 Nat Commun https://doi.org/10.1038/s41467-021-26690-1; Maurer et al., 2022 Stem Cell Research, https://doi.org/10.1016/j.scr.2022.102677) with the minor variation of differentiating iPSCs towards atrial iPS-CMs, which does not provide sufficient novelty to the data.  

Additional comments

1.     Fig 1a showing the sequences of SCN10A knockout iPSC lines seems to been already used in Supplementary Fig. 5 of Bengel et al., 2021 Nat Commun https://doi.org/10.1038/s41467-021-26690-1 and Fig. 1 of Maurer et al., 2022 Stem Cell Research, https://doi.org/10.1016/j.scr.2022.102677. Citing the original paper describing the generation of the SCN10A knockout iPSC lines should be sufficient.

2.     Authors wrote (lines 319-321) “Our data show. Since dv/dt is a surrogate for the fast Na+ influx and peak Na+ current, these data show that there is no involvement of NaV1.8 on the peak Na+ current in atrial iPSC CMs.” Please correct the sentence and add the missing maximal upstroke velocity data (dv/dt max) to the Figure 3. These data are critical to estimate the effect of PF-01247324 treatment to the peak sodium current.

3. The stimulation frequency should be clearly indicated for the AP measurements (Fig 3).

4.     Please present also APD20, APD50 and APD70 metrics for thorough estimation of action potential changes.

5. The cell plating density for calcium sparks and calcium handling analyses is not specified. Were these measurements done in single cells or cell clusters? The temperature conditions for calcium measurements are not specified.

6.     The references should be corrected (lines 54-55, 425).

7.     The image quality of Fig 5 should be improved.

Author Response

Reviewer 1:

The current study utilized SCN10A knockout atrial iPSC-CMs to investigate the role of the Nav1.8 isoform (encoded by SCN10A gene) in the electrophysiological properties of CMs and assess the potential of it inhibition as a clinical target for the treatment of atrial arrhythmias.

Authors found that genetic knockout and pharmacological inhibition of NaV1.8 significantly reduced INaL in atrial iPSC-CM, and the specific NaV1.8 inhibitor reduced INaL in control iPSC-CM to the level of KO iPSC-CM. Whole-cell current-clamp experiments were performed to evaluate the effects of genetic KO and pharmacological inhibition of NaV1.8 on action potential parameters in human atrial iPSC-CM, and no significant effects were observed on action potential duration, resting membrane potential, or action potential amplitude. The study found that both knockout of SCN10A and pharmacological inhibition of NaV1.8 led to a decrease in spontaneous arrhythmogenic Ca2+ sparks and waves in human atrial iPSC-CM, indicating a potential role of NaV1.8-dependent INaL in cellular arrhythmogenesis. Fura 2-AM calcium measurements demonstrated intact Ca2+ handling in both atrial iPSC-CM with and without KO of SCN10A and/or pharmacological inhibition of NaV1.8, with no significant effects on Ca2+ transient parameters.

Despite the clear relevance of the topic, the current study lacks originality and novelty and therefore cannot be recommended for publication. The authors used to investigate extensively the role of Nav1.8 in various models, including human heart failure atrial CMs (Dybkova, et al., 2018 Cardiovascular Research https://doi.org/10.1093/cvr/cvy152), wild type and SCN10A knockout murine CMs, and wild type and SCN10A knockout human ventricular iPSC-CMs (Bengel et al., 2021 Nat Commun https://doi.org/10.1038/s41467-021-26690-1). The present study uses previously generated SCN10A knockout iPSC lines (Bengel et al., 2021 Nat Commun https://doi.org/10.1038/s41467-021-26690-1; Maurer et al., 2022 Stem Cell Research, https://doi.org/10.1016/j.scr.2022.102677) with the minor variation of differentiating iPSCs towards atrial iPS-CMs, which does not provide sufficient novelty to the data.  

Answer: We thank you for the detailed analysis of our data and agree in principle. However, we see a clear novelty in our data, of which we would like to convince the reviewer.

From a clinical point of view, the present work underlines and complements the previously published data, particularly about atrial arrhythmogenesis. However, patients with persistent or long-standing atrial fibrillation or heart failure patients characterized by advanced structural atrial remodeling are unlikely to be optimal patients for a pharmacological rhythm strategy. Since many antiarrhythmogenic drugs have severe side-effects or have not been approved to use, other antiarrhythmic strategies are needed In these patients in particular, the CAST TRIAL has shown that drug therapy with class 1 A antiarrhythmic drugs may lead to more atrial arrhythmias and more deaths due to arrhythmias. Furthermore, atrial arrhythmias may have a negative impact on the progression of heart failure, especially in these patients. Therefore, it is critical to expand the variety of therapies that lead to a reduction in atrial arrhythmias, including a thorough investigation of atrial electrophysiology. Therefore, extending our experiments to human atria using atrial iPSC-CM provides a thorough investigation and solution to a clinically unmet need. The studies published so far were limited by the use of inhibitors, as these ion channel blockers may also have nonspecific side effects. Therefore, we performed the ultimate proof-of-principle approach with the CRISPR-Cas9 method using a SCN10A knockout in atrial iPSC-CM. Here, we extend the previously shown data by performing experiments in human atrial cells, outside the previously mentioned mouse models, and can additionally selectively determine the proportion of INaL by NaV1.8.

To date, only INaL has been measured in our ventricular iPSC-CM model. In the present manuscript, we provide detailed experimental/electrophysiological studies of atrial cells with AP measurements and kinetics, calcium balance, and leak, which has not been shown previously in either ventricular or atrial iPSC-CM. We believe that the functional evidence presented here and, in particular, the potent antiarrhythmic effects of inhibition and deletion of NaV1.8 are of fundamental importance for the development of new therapeutic strategies for atrial arrhythmias.

Additional comments

  1. Fig 1a showing the sequences of SCN10A knockout iPSC lines seems to been already used in Supplementary Fig. 5 of Bengel et al., 2021 Nat Commun https://doi.org/10.1038/s41467-021-26690-1 and Fig. 1 of Maurer et al., 2022 Stem Cell Research, https://doi.org/10.1016/j.scr.2022.102677. Citing the original paper describing the generation of the SCN10A knockout iPSC lines should be sufficient.

Answer: We thank the reviewer for this important comment and agree that the sequences of the different NaV1.8-KO-iPSC lines (62.1 and 62.4) have already been shown in Maurer et al. 2022 and Bengel et al. 2021. Although data from other sequencing runs of the same iPSC lines were used in this manuscript, we have removed the Nav1.8-KO-iPSC line in Figure 1a and now focus on sequencing of the atrial iPSC cardiomyocytes, which have not been shown anywhere previously. The entire Figure 1 now describes atrial Nav1.8-KO-iPSC cardiomyocytes compared with control cells.

Figure 1. CRISPR/Cas9 based knock-out of SCN10A/NaV1.8 in atrial iPSC-CM. (a) Sanger sequencing of control iPSC, and SCN10A iPSC-KO cardiomyocytes (CM) demonstrating frameshifts in both alleles leading to premature stop in exon 1 (A1: delC/insCAC and A2: delCT) (b) Atrial control and SCN10A KO iPSC-CM were stained for MLC2a (green), and MLC2v (red) demonstrating atrial differentiation. Nuclei were stained with DAPI. (c) mRNA expression level of atrial marker PITX2 normalized to house-keeping gene HPRT in atrial control and SCN10A KO iPSC-CM (n=6 vs. 3 differentiations) compared to ventricular control and SCN10A KO iPSC-CM (n=7/4 differentiations).

  1. Authors wrote (lines319-321) “Our data show. Since dv/dt is a surrogate for the fast Na+ influx and peak Na+ current, these data show that there is no involvement of NaV1.8 on the peak Na+ current in atrial iPSC CMs.” Please correct the sentence and add the missing maximal upstroke velocity data (dv/dt max) to the Figure 3. These data are critical to estimate the effect of PF-01247324 treatment to the peak sodium current.

Answer: We fully agree with the reviewer that this has to be demonstrated. We took these criticisms seriously, corrects the sentence and added/changed the following text: “No significant effects of KO or pharmacological inhibition of NaV1.8 on either AP amplitude (APA, Figure 3c, 113.7±4.4 ms, vs. control + PF 118.7±3.4 ms, KO control 105.7±5.1 ms, KO + PF 102.1±4.2 ms, ); resting membrane potential (RMP , Figure 3d, -76.0±6.2 ms, vs. control + PF -72.2±5.7 ms, KO control -66.7±2.6 ms, vs KO + PF -67.3±3.9 ms) or upstroke velocity (Vmax, Figure 3e, 106.2±11.2 vs. control + PF 127.5±11.9 mV/ms, KO control 92.7±13.8 vs. KO + PF 82.8±12.4 mV/ms) could be observed.” (page 7, line 178-179). Moreover, we added new data (Figure 3e) showing the maximal upstroke velocity data.

Figure 3: Mean data ± SEM of Vmax (atrial control n=18 cells/4 differentiations; atrial control + PF n=22 cells/5 differentiations; SCN10A KO control n=19 cells/5 differentiations, SCN10A KO + PF n=19 cells/4 differentiations).

  1. The stimulation frequency should be clearly indicated for the AP measurements (Fig 3).

Answer: As suggested by reviewer we added the following text: “ The data presented herein are representative of measurements conducted at a frequency of 1 Hz.” (page 7, line 163-164).

  1. Please present also APD20, APD50 and APD70 metrics for thorough estimation of action potential changes.

Answer: We thank the reviewer for this constructive comment. Furthermore, no discernible impacts were observed on the duration of atrial action potential at 20% repolarization (APD20), action potential duration at 50% repolarization (APD50), and action potential duration at 70% repolarization (APD70). The available data, including table S1 and figure S1, were included in the supplemental material.  We now provide the requested data with APD20, APD50 and APD70 of our experiments. The table and the figure below were added to the supplement as supplementary table S1 and figure S1. We included the description in the supplement and the results (page 7, line 168-172).

APD20

APD50

APD70

atrial control

34.2±7.9

77.2±13.1

109.8±15.9

atrial control + PF

35.1±7.6

82.3±12.4

118.9±14.5

SCN10A KO control

32.0±7.0

72.0±12.5

109.8±17.2

SCN10A KO + PF

14.5±5.7

63.7±15.7

90.1±18.2

Table S1:  Action potential duration of atrial iPSC-CM (APD20, APD50 and APD70) at 1 Hz. Mean data (nested) ± SEM of APD20 (atrial control n=18 cells/4 differentiations; atrial control + PF n=22 cells/5 differentiations; SCN10A KO control n=19 cells/5 differentiations, SCN10A KO + PF n=19 cells/4 differentiations); (b) Mean data (nested) ± SEM of APD50 (atrial control n=18 cells/4 differentiations; atrial control + PF n=22 cells/5 differentiations; SCN10A KO control n=19 cells/5 differentiations, SCN10A KO + PF n=19 cells/4 differentiations) and (c) Mean data (nested) ± SEM of APD70 (atrial control n=18 cells/4 differentiations; atrial control + PF n=22 cells/5 differentiations; SCN10A KO control n=19 cells/5 differentiations, SCN10A KO + PF n=19 cells/4 differentiations); statistics with nested 1 way ANOVA.

Figure S1: (a) Mean data (nested) ± SEM of APD20 (atrial control n=18 cells/4 differentiations; atrial control + PF n=22 cells/5 differentiations; SCN10A KO control n=19 cells/5 differentiations, SCN10A KO + PF n=19 cells/4 differentiations); (b) Mean data (nested) ± SEM of APD50 (atrial control n=18 cells/4 differentiations; atrial control + PF n=22 cells/5 differentiations; SCN10A KO control n=19 cells/5 differentiations, SCN10A KO + PF n=19 cells/4 differentiations) and (c) Mean data (nested) ± SEM of APD70 (atrial control n=18 cells/4 differentiations; atrial control + PF n=22 cells/5 differentiations; SCN10A KO control n=19 cells/5 differentiations, SCN10A KO + PF n=19 cells/4 differentiations); statistics with nested 1 way ANOVA.

  1. The cell plating density for calcium sparks and calcium handling analyses is not specified. Were these measurements done in single cells or cell clusters? The temperature conditions for calcium measurements are not specified.

Answer: We apologize that the information of the research design is inadequate. We added the details of cell number and temperature conditions in the manuscript:  35.000 atrial iPSC-CM were plated on glass-bottom Fluoro Dishes and incubated with either isoprenaline (50 nmol/L, Sigma) or isoprenaline + PF01247324 (1 µmol/L, Sigma) for 15 min before starting measurements. Experiments were conducted at room temperature. (page 20, line 485-489). And “35.000 atrial iPSC-CM” were added at (page 21, line 537 + line 551).

  1. The references should be corrected (lines 54-55, 425).

Answer: The mentioned references were corrected.

  1. The image quality of Fig 5 should be improved.

Answer: We completely agree with the reviewer and carefully revised the representative images.

Reviewer 2 Report

The manuscript by Hartmann analyzed the contribution of SCN10A/Nav1.8 to INaL using a gene edited model of human iPSC differentiated into atrial cardiomyocytes. The authors nicely showed that selective gene editing SCN10A leads to changes in INaL, an effect that mimics the results obtained by using selective Nav1.8 blockers. Furthermore, the authors demonstrated that no changes in the action potential were observed (APD90) while selective changes in calcium spark were recorded. Overall the study is quite interesting, further support the functional contribution of SCN10A/Nav1.8 to INaL current and thus to arrhythmogenesis. However, there are several issues that remain uncleared

First of all, I wouldn´t say is that the present mode is a knockout, since there is residual expression of Scn10a/Nav1.8. Furthermore, the fact that in the sequencing chromatograms there is overlapping signals from the Crispr/Cas9 editing point, means that there are co-existance of wild-type and mutated cells. Thus, I would say that these cell lines represent a knockdown but not knockout models.

Secondly, I am surprised that no changes in the duration of the action potential are observed. How do you reconcile this findings, as if INaL was prolonged (i.e. controls), the action potential is expected to be longer. I am missing something?

Thirdly, Figure 6 is missing. It is compulsory to show this data since changes in SCN5A/Nav1.5 are critical to fully ascertain that changes observed in this study are genuine and exclusively due to SCN10A deletion, taking into account the gene editing strategy might interfere with SCN5A expression, since  SCN5A // SCN10A are clustered in the human genome very tightly.

Author Response

Reviewer 2:

Comments and Suggestions for Authors

The manuscript by Hartmann analyzed the contribution of SCN10A/Nav1.8 to INaL using a gene edited model of human iPSC differentiated into atrial cardiomyocytes. The authors nicely showed that selective gene editing SCN10A leads to changes in INaL, an effect that mimics the results obtained by using selective Nav1.8 blockers. Furthermore, the authors demonstrated that no changes in the action potential were observed (APD90) while selective changes in calcium spark were recorded. Overall the study is quite interesting, further support the functional contribution of SCN10A/Nav1.8 to INaL current and thus to arrhythmogenesis. However, there are several issues that remain uncleared.

Answer: We thank you for appreciating our work and the precise help in order to improve the manuscript.

First of all, I wouldn´t say is that the present mode is a knockout, since there is residual expression of Scn10a/Nav1.8. Furthermore, the fact that in the sequencing chromatograms there is overlapping signals from the Crispr/Cas9 editing point, means that there are co-existance of wild-type and mutated cells. Thus, I would say that these cell lines represent a knockdown but not knockout models.

Answer: We thank the reviewer for this question/comment. However, the answer to this question is rather complex and hence, we tried to approach this topic extensively in agenetic as well as protein level. Identifying the precise editing mechanisms in the generated homozygous knock-out clones is challenging. Analysis of the actual sequencing result indeed was difficult because spontaneous and unpredictable insertions and deletions were intended to occur to cause premature stop codons during this editing procedure. Because of this, in addition to visual inspection, the online tool CRISP-ID (http://crispid.gbiomed.kuleuven.be/) was employed. Based on this analysis of the sequencing results, it is possible to explain the overlapping signals in the sequencing chromatograms by contrasting editing mechanisms on the two different alleles, as shown in Fig. 1, and not the presence of the wildtype allele. When the sequencing traces for the two different alleles are separated, it can be concluded that insertions and/or deletions occurred on both alleles (cytosin was deleted and CAC trinucleotide was inserted on the first allele and a CT dinucleotide was deleted on the second allele) which results in early stop codons in the first exon on both alleles. It is no longer possible to identify the entire wildtype sequence in these sequencing chromatograms.

Cell clones with just one altered allele and one wt allele (heterozygous knock out) were also created during the editing process. These clones however were not included in the experiments of this manuscript. But to make things clearer, we included a heterozygous gene edited iPSC clone in the answers for the reviewer in comparison to the homozygous SCN10A iPSC-lines that were used in our study:

The ‘residual expression’ in the Western Blots appear in some differentiation experiments at higher kDa sizes than Nav1.8, but in most experiments no bands were detected in the KO-lines at the expected size for Nav1.8. We quantified again (specifically the bands at the right size) and included more cardiac iPSC differentiations to clarify this point. For the functional analyses, we used almost exclusively the blue (crtl) and red bordered (Nav1.8 -KO) iPSC-CM.

In conclusion, on a genetic level, insertions and/or deletions occurred on both alleles of SCN10A leading to multiple stop codons in Exon 1. No wt allele is left. These genetic data strongly point to a knock out. On protein expression level using Western Blot technique, we found significantly decreased Nav1.8 in the gene edited iPSC-cardiomyocytes. Based on genetic and protein analysis we conclude to have a knock out instead of a knockdown and that residual signals are from the polyclonality of the used antibody for Nav1.8 at higher kDa sizes. We used only Nav1.8 KO differentiation for functional analysis, in which no bands or almost no bands (and these ran at the wrong higher sizes) were detected in Western blots.

To take the reviewer's advice seriously, we have removed the Western Blots from the manuscript (Figure 1) and have only shown them in the answers. However, these can be put back in if desired.

Secondly, I am surprised that no changes in the duration of the action potential are observed. How do you reconcile this findings, as if INaL was prolonged (i.e. controls), the action potential is expected to be longer. I am missing something?

Answer:  We fully agree with the reviewer's statement, as it is indeed accurate. The human atrial cardiomyocytes generated from induced pluripotent stem cells (iPSC-CM) used in our study were derived solely from healthy individuals, where the magnitude of the late sodium current (INaL) is considered to be small. However, under conditions where INaL is enhanced, such as chronic CaMKIIδc overexpression [3], heart failure [3,4], or other pathological conditions, clear effects on action potential duration (APD) prolongation were observed. This observation is consistent with our previous investigations in the human atria [2]. Furthermore, the interaction of NaV1.8, for example, with enhanced CaMKIIδc activity, may be necessary to generate meaningful effects on cardiomyocyte electrophysiology, specifically APD. It is important to note that NaV1.8 appears to play a negligible role in healthy cardiomyocytes. This establishes NaV1.8 as a disease-specific target. Although INaL is not significantly increased in healthy cardiomyocytes, we were able to demonstrate continuous modulation of Na/Ca, as evidenced by the results of calcium sparks, which showed clinically relevant antiarrhythmic effects.

Thirdly, Figure 6 is missing. It is compulsory to show this data since changes in SCN5A/Nav1.5 are critical to fully ascertain that changes observed in this study are genuine and exclusively due to SCN10A deletion, taking into account the gene editing strategy might interfere with SCN5A expression, since  SCN5A // SCN10A are clustered in the human genome very tightly.

Answer: We thank the reviewer for this important comment. To demonstrate that NaV1.5 is not affected in atrial iPSC-cardiomyocytes, we performed Western-Blot experiments from Ctrl and SCN10A-Knock-Out cells. As new figure 6 above indicates, there was no significant effect of SCN10A knock-out on NaV1.5 protein expression demonstrating that the changes observed in this study are indeed and exclusively due to deletion of SCN10A. Finally, to take the concern serious we added data (figure 6 a-f) with expression level of CaV1.2, and RyR2 protein in atrial control and SCN10A KO iPSC-CM normalized to GAPDH. (page 16, line 283-289).

Figure 6: Original Western blots of NaV1.5 (ad), CaV1.2 (fb) and RyR2 (gc) in atrial control and SCN10A KO iPSC-CM. Normalized values of NaV1.5 (a) n=3 control/6 KO differentiations), CaV1.2 (b) n=5/7 differ-entiations), RyR2 (c) n=5/7 differentiations in atrial control and SCN10A KO iPSC-CM normalized to GAPDH (n=5/7 differentiations).

In addition, we analyzed whether the gene editing strategy for SCN10A KO interferes with SCN5A by using specific programs. We were able to identify the specific off targets of the used guide RNAs for SCN10A KO and demonstrated that SCN5A was no predicted off-target for the Cas nuclease activity for the two used guide RNAs 1 and 2. Nevertheless, we demonstrated the lack of NHEJ-caused mutagenesis in the top predicted off-target Cas nuclease activity for gRNA1 (ADORA3, CACNA2D4, KCTN1, MDH2) and for gRNA2 (CRB2, SCG5, SCL39A11, SNCA) (Maurer et al., 2022).

In conclusion, it can be assumed that NaV1.5 expression and genomic integrity is not influenced by SCN10A KO in iPSC-CM.

  1. Jabbari, J.; Olesen, M.S.; Yuan, L.; Nielsen, J.B.; Liang, B.; Macri, V.; Christophersen, I.E.; Nielsen, N.; Sajadieh, A.; Ellinor, P.T.; et al. Common and rare variants in SCN10A modulate the risk of atrial fibrillation. Circ Cardiovasc Genet 2015, 8, 64-73, doi:10.1161/HCG.0000000000000022.
  2. Pabel, S.; Ahmad, S.; Tirilomis, P.; Stehle, T.; Mustroph, J.; Knierim, M.; Dybkova, N.; Bengel, P.; Holzamer, A.; Hilker, M.; et al. Inhibition of NaV1.8 prevents atrial arrhythmogenesis in human and mice. Basic Res Cardiol 2020, 115, 20, doi:10.1007/s00395-020-0780-8.
  3. Bengel, P.; Dybkova, N.; Tirilomis, P.; Ahmad, S.; Hartmann, N.; B, A.M.; Krekeler, M.C.; Maurer, W.; Pabel, S.; Trum, M.; et al. Detrimental proarrhythmogenic interaction of Ca(2+)/calmodulin-dependent protein kinase II and NaV1.8 in heart failure. Nat Commun 2021, 12, 6586, doi:10.1038/s41467-021-26690-1.
  4. Dybkova, N.; Ahmad, S.; Pabel, S.; Tirilomis, P.; Hartmann, N.; Fischer, T.H.; Bengel, P.; Tirilomis, T.; Ljubojevic, S.; Renner, A.; et al. Differential regulation of sodium channels as a novel proarrhythmic mechanism in the human failing heart. Cardiovascular research 2018, 114, 1728-1737, doi:10.1093/cvr/cvy152.

Reviewer 3 Report

The manuscript describes that Nav1.8 (SCN10A) may specifically involved in the late Na current and diastolic SR Ca2+ leak, which are associated with arrythmogenesis. Other parameters were not altered significantly by SCN10A-/- nor inhibitors. The authors used CRISPR/Cas9 to make SCN10A-/- iPSCs. Therefore, the genetic background difference should be least. The results and conclusions were adequately suggested by the experiments. The manuscript is very interesting, and basically acceptable.

A few minor revisions may be necessary.

Lines 272-273, the meaning of “important” is difficult to understand. Important for keeping pathological condition? Clarify this sentence.

Line 289, why “in contrast”? The authors’ present study could detect very low INaL, but Casini did not et al.? However, Casini detected it after beta-adrenergic stimulation, meaning there can be low INaL. I think it is better to delete in contrast, or put it in other words.

Line 55, Line 426, what do # and reference mean?

Author Response

Reviewer 3

Comments and Suggestions for Authors

The manuscript describes that Nav1.8 (SCN10A) may specifically involved in the late Na current and diastolic SR Ca2+ leak, which are associated with arrythmogenesis. Other parameters were not altered significantly by SCN10A-/- nor inhibitors. The authors used CRISPR/Cas9 to make SCN10A-/- iPSCs. Therefore, the genetic background difference should be least. The results and conclusions were adequately suggested by the experiments. The manuscript is very interesting, and basically acceptable.

Answer:  We would like to thank the reviewer for these kind comments.

A few minor revisions may be necessary.

Lines 272-273, the meaning of “important” is difficult to understand. Important for keeping pathological condition? Clarify this sentence.

Answer: We are sorry for not being clear enough. We modified in the revised manuscript:  “Under pathological conditions, the enhanced persistent Na+ influx, known as enhanced INaL, has been demonstrated to play an important role throughout the action potential.” (page 17, line 335-337).

Line 289, why “in contrast”? The authors’ present study could detect very low INaL, but Casini did not et al.? However, Casini detected it after beta-adrenergic stimulation, meaning there can be low INaL. I think it is better to delete in contrast, or put it in other words.

Answer:   As suggested by reviewer we modified/delete in the revised manuscript “in contrast”. (page 18, line 356)

Line 55, Line 426, what do # and reference mean?

Answer: We apologize for not presenting a perfect high standard manuscript. The mentioned references were corrected.

Reviewer 4 Report

The study describes the generation of human atrial cardiomyocytes with homozygous knockout of the SCN10A gene, which codes for the voltage-gated sodium channel NaV1.8. The researchers used CRISPR/Cas9 genome editing to generate the knockout cells and confirmed the loss of NaV1.8 protein expression by Western blot. The study shows that the reduction of proarrhythmogenic late sodium current (INaL) is significant in human atrial cardiomyocytes with the SCN10A knockout. However, there were no effects observed on atrial action potential characteristics, resting membrane potential, or action potential amplitude in the SCN10A knockout cells. The study also suggests that NaV1.8 exerts its arrhythmogenic potential in the atria by enhancing INaL.

The paper appears to be written in a clear and concise manner. The authors provide a clear description of the experimental methods used to investigate the role of NaV1.8 in atrial CM and present their findings in a logical and well-organized manner.

Author Response

Reviewer 4:

Comments and Suggestions for Authors

The study describes the generation of human atrial cardiomyocytes with homozygous knockout of the SCN10A gene, which codes for the voltage-gated sodium channel NaV1.8. The researchers used CRISPR/Cas9 genome editing to generate the knockout cells and confirmed the loss of NaV1.8 protein expression by Western blot. The study shows that the reduction of proarrhythmogenic late sodium current (INaL) is significant in human atrial cardiomyocytes with the SCN10A knockout. However, there were no effects observed on atrial action potential characteristics, resting membrane potential, or action potential amplitude in the SCN10A knockout cells. The study also suggests that NaV1.8 exerts its arrhythmogenic potential in the atria by enhancing INaL.

The paper appears to be written in a clear and concise manner. The authors provide a clear description of the experimental methods used to investigate the role of NaV1.8 in atrial CM and present their findings in a logical and well-organized manner.

Answer: We appreciate these encouraging comments and the Reviewer’s assessment of our investigations.

Round 2

Reviewer 1 Report

After carefully reviewing the revised manuscript, I regret to say that changes made by authors are not sufficient to recommend the manuscript for publication. I appreciate that the authors have made an effort to address my previous comments, but the minor revisions made do not address the fundamental issues. The experiments presented in the revised manuscript lack the necessary depth to provide the required novelty and significance.

As mentioned earlier, the authors have previously published a demonstration of the direct association between SCN10A and INaL in a human context, specifically in ventricular iPSC-CMs. 

Other electrophysiological measurements have not been previously reported, but the data presented in a confusing manner, which could potentially lead to incorrect conclusions. Averaging of single cell measurements from the same differentiation leads to a reduction in replicate numbers, resulting in a loss of statistical power. Additionally, the measured value distribution could not be properly evaluated in this case. Each cell measured should be plotted on the graphs and used for statistical comparisons between conditions. The AP experiments and calcium handling experiments should be critically re-analysed. 

For example, despite the authors' claim that "data show that there is no involvement of NaV1.8 on the peak Na+ current in atrial iPSC CMs," it appears that Vmax (and APD90 to a smaller extent) is decreased in SCN10A KO groups. This observation is important and should be clearly presented with peak sodium current measurements if necessary.

Experiments evaluating calcium sparks seem to be solid. However, it is not clear whether the calcium sparks generation mechanism in WT iPSC-CMs could reproduce those associated with atrial fibrillation or if it is simply a reflection of the immaturity of the cell model.

The use of iPSC-CMs from patients with atrial fibrillation would help to address this question and would bring missing novelty valuable insights.